# Laparoscopy in Gynecologic and Abdominal Surgery in Regional (Spinal, Peridural) Anesthesia, the Utility of the Technique during COVID-19 Pandemic

**DOI:** 10.3390/medicines8100060

**Published:** 2021-10-19

**Authors:** Attila Louis Major, Kudrat Jumaniyazov, Shahnoza Yusupova, Ruslan Jabbarov, Olimjon Saidmamatov, Ivanna Mayboroda-Major

**Affiliations:** 1Femina Gynecology Centre, CH-1205 Geneva, Switzerland; 2Department of Obstetrics & Gynecology, University of Fribourg, CH-1700 Fribourg, Switzerland; 3Department of Obstetrics and Gynecology, Urgench Branch of Tashkent Medical Academy, Urgench 220100, Uzbekistan; kudratulla@mail.ru (K.J.); shahnoza.yusupova90@gmail.com (S.Y.); ruslonzabborovl@gmail.com (R.J.); 4Faculty of Tourism and Economics, Urgench State University, Urgench 220100, Uzbekistan; 5Department of Gynecology and Obstetrics, University Hospital of Geneva, CH-1205 Geneva, Switzerland

**Keywords:** laparoscopy, surgery, gynecology, general anesthesia, regional anesthesia, spinal anesthesia, COVID-19 complications

## Abstract

Background: laparoscopic surgery is mainly performed in general anesthesia. Symptomatic patients infected with COVID-19 needing surgery are however at higher risk for COVID-19 complications in general anesthesia than in regional anesthesia. Even so, Covid transfection is a hazard to medical personnel during the intubation procedure and treatment drugs may be in shortage during a pandemic. Recovery and hospital stay are also shorter after laparoscopy. Laparoscopy performed in regional anesthesia may have several advantages in limiting Covid. Methods: international literature on the risk of COVID-19 complications development was searched. 3 topics concerning laparoscopic surgery were reviewed: (1) Achievements in laparoscopy; (2) Advantages of regional anesthesia compared to general anesthesia; (3) Feasibility to perform laparoscopy in regional anesthesia in COVID-19 pandemic. The authors reviewed abstracts and full-text articles concerning laparoscopic surgery, gynecology, anesthesia and COVID-19. Studies published in PubMed, Embase, Cochrane Library and found in Google Scholar before 1st FEB, 2021 were retrieved and analyzed. Results: a total of 83 studies were found, all of them written in English. 17 studies could be found in gynecology and in general surgery about laparoscopy with regional anesthesia. In Covid time only one study compared laparoscopic surgery in general anesthesia to laparotomy and another study laparotomy in general anesthesia to regional anesthesia. Laparoscopy showed no disadvantage compared to laparotomy in Covid pandemic and in another study laparotomy in general anesthesia was associated with higher mortality and more pulmonary complications. Trendelenburg position can be a threat if used by inexperienced personnel and can induce unintended anesthesia of breathing organs. On the other hand Trendelenburg position has advantages for cardiovascular and pulmonary functions. Pneumoperitoneum of low CO_2_ pressure is well tolerated by patients. Conclusions: elective surgery should be postponed in symptomatic Covid patients. In inevitable emergency surgery intubation anesthesia in COVID-19 pandemic is as far as possible to be avoided. In COVID-19 pandemic, regional anesthesia is the preferred choice. The optimum may be the combination of laparoscopic surgery with regional anesthesia. Reducing the pneumoperitoneum is a good compromise for the comfort of patients and surgeons. A special case is gynecology, which needs to be performed in Trendelenburg position to free pelvic organs.

## 1. Introduction

Laparoscopic surgery became the first choice in many indications of abdominal and gynecological surgery due to less invasiveness and better vision and shorter recovery time. The standard for laparoscopy is general anesthesia. However, due to complications caused by Covid infection it may be preferable to avoid intubation. Since the transmission of Covid to medical personnel is increased in an emergency situation, regional anesthesia such as spinal or peridural anesthesia should be favored if possible [1,2]. Regional anesthesia is applied as routine in abdominal surgery with a lower abdominal horizontal incision. An example for laparotomy in regional anesthesia is cesarean section and for vaginal surgery hysterectomy. Some emergency surgeries such as extrauterine pregnancy or appendicitis are carried out only by the abdominal route. In many centers laparoscopic surgery in general anesthesia became the standard. Moreover, gynecological laparoscopic surgery is carried out in Trendelenburg positions to liberate pelvic organs for a better view. The feasibility of laparoscopic surgery in regional anesthesia was demonstrated, mainly for cholecystectomies and pelvic surgeries [2].

Patients infected with COVID-19 needing surgery are at risk for COVID-19 complications in general anesthesia in contrast to regional anesthesia. Proved long-term complications of COVID-19 (3 months–1 year) were reported to be: fibrosis of the lungs, bronchitis, chronic pulmonary failure, myocarditis, arrhythmias and chronic heart failure, liver and kidney damage, demyelination of nerve fibers, cognitive impairment, depression and psychosis. For the practice of regional anesthesia during the COVID-19 pandemic Uppal et al. published an interim joint statement by the American Society of Regional Anesthesia and Pain Medicine and the European Society of Regional Anaesthesia and Pain Therapy [1,2]. This paper provides evidence-based practice recommendations for the safe performance of regional anesthesia during COVID-19 pandemic. It is recommended that regional anesthesia should be preferred to general anesthesia in COVID pandemic, if there are no contraindications [3]. In a systematic review, the incidence of postoperative pneumonia in patients undergoing spinal or epidural anesthesia was lower (odds ratio, 0.63) than that in patients undergoing general anesthesia [4,5]. 

The aim of this review is to analyze the studies on laparoscopy in regional anesthesia and to take some conclusions concerning its utility in Covid pandemic.

### 1.1. Achievements in Laparoscopy

Laparoscopy started in the mid-1950s when gynecologists discovered this technique as a safe method [6,7] to diagnose pelvic pain while decreasing hospital stay and pain after operation [8,9]. Thus, laparoscopy for general surgery purposes scientifically demonstrated to be advantageous in reduction of postoperative pain [10], less intraoperative bleeding, better cosmetic performance, quicker return to normal lifestyle, reduction in hospital stay leading to overall reduction in medical cost [11], less postoperative pulmonary complications, less postoperative wound infection, less metabolic derangement [12,13] and better postoperative respiratory function [14].

Technological advancements in optics, illumination, video technology and instrumentation have further expanded the frontiers from diagnostic to operative laparoscopy. As far as the list of laparoscopic procedures extends, most of the conventional abdominal or pelvic surgical procedures can be performed via a minimal invasive approach using the laparoscope. State-of-the-art laparoscopic surgery is continuously progressing and has become the mainstay of operative management in gynecology and surgery in centers with the adequate equipment and training. Nowadays, laparoscopic surgeries such as tubal ligation, adnexal surgery, myomectomy, hysterectomy, appendicectomy, cholecystectomy, surgery on the intestines and even cases of cancer are becoming reliable alternative to classic laparotomies. The shorter hospital stay and quick recovery period have a positive effect on a patient’s quality of life, as they return to a regular lifestyle quickly [15].

Laparoscopic processes are often performed under general anesthesia (GA) using endotracheal intubation to prevent aspiration, respiratory distress, discomfort and shoulder pain because of induction of pneumoperitoneum. The application of loco-regional anesthesia (LRA) during laparoscopy is usually unified with general anesthesia (GA) towards reducing postoperative pain [16,17]. For cholecystectomies and surgeries on the uterine adnexa more and more studies are showing the feasibility and advantages of laparoscopic surgeries carried out in single regional anesthesia. Under certain conditions, such as in patients with severe comorbidities limiting GA, LRA is used as the exclusive anesthesiology method [16]. This was reported for more important surgery as laparoscopic hysterectomy performed under regional anesthesia in patients with chronic obstructive pulmonary disease [16].

### 1.2. Regional Anesthesia: More Than Acute Pain Management

Regional anesthesia has faced an increasing improvement in the last few decades [17]. Regional anesthesia has consistent function to minimize perioperative opioid requirements, decrease in patient mortality, major morbidity (e.g., pulmonary complications, transfusion requirements) and economic benefits such as length of hospital stay [18].

The most serious respiratory syndrome coronavirus (SARS-CoV) pandemic was a reason to unprecedented proportions and significantly impacted on healthcare facilities and surgical volume [19]. Anesthesia is needed for acute and emergency surgery. General anesthesia with airway intervention accelerates aerosol generation, which exposes the healthcare team to risk of dissemination of COVID-19 both during tracheal intubation and extubation [20].

Similar to previous pandemics, healthcare professionals are unprotected against infections. Thus, strategies directed to reduce exposure and the risk of disease transmission to medical personnel or patients in the hospital is very crucial. Acute respiratory infection during tracheal intubation to a medicine personnel are proved to be 6.6 times compared to anesthesia without intubation [21].

Regional anesthesia is correlated with a low risk of postoperative complications that makes it essential in the context of continuing respiratory infection. Due to acute COVID -19 pneumonia many patients develop complications such as fibrosis of the lungs, bronchitis, chronic pulmonary failure, oxygen starvation, myocarditis, arrhythmia and chronic heart failure, liver and kidney damage, cognitive impairment, depression and psychosis [22,23]. In order not to favor such complications general anesthesia during elective surgery is better to be avoided during the pandemic period, if additional material such as filters and trained personnel is not available. In addition, the use of large doses of antibiotics and hormonal drugs leads to the appearance of mycotic flora and a decrease in immunity. As a consequence, the damage of the upper respiratory tract during intubation and extubation, can lead to severe nosocomial secondary pneumonia. Regional anesthesia may be the preferred choice under these circumstances [24].

The potential benefit of perioperative regional anesthesia is more than acute pain relief. Numerous retrospective research on clinical databases have revealed endoscopic surgery in neuraxial anesthesia such as spinal anesthesia among others to be correlated with a reduction in patient mortality and major morbidity such as pulmonary complications and transfusion requirements, compared with classical surgery in general anesthesia. In addition, economic parameters such as duration of hospital stay were also decreased [17].

Better postoperative pain control can be partly linked with the spinal drug injection in the LRA group. Instead, when GA patients rejoined consciousness postoperatively, they were encountering initial pain. To compare the advantages of SA on postoperative pain with GA maintained statistical importance at subsequent testing at 8, 12, 24 and 48 h in the postoperative period. No woman in the SA group required extra intravenous opioid administration. Foremost, they accomplished quick return of bowel function and independent de-ambulation. The quick recovery in the postoperative period allows short time of urinary catheter and decreases the risk of infection. Reasons contributing to better postoperative pain control are to avoid a longer bed stay, which can cause the appearance of paralytic ileus, muscular pain and fatigue [25]. Intra- and post-operative advantages of SA combined with LRA were observed. Significantly lower pain scores were reported compared to GA in the postoperative period. The significant difference was detected one hour after surgery and a statistically major 6-points difference in visual analog scale pain scoring (VAS) was recorded [25]. VAS is a validated, subjective measurement of acute and chronic pain, which we are using for our own study in laparoscopic surgery in SA.

### 1.3. Local Anesthesia

Current technological advancement in optical fiber technology has assembled laparoscopes with exterior diameters of as small as 1.2 to 2.2 mm. These devices give the opportunity to perform ‘‘micro- laparoscopy” with local anesthesia solely or supplemented by sedation. With such technology local anesthesia can be applied as a safe, definitive and economical alternative to general anesthesia. It is secure, efficient and less expensive and has been mostly used for patients with infertility, chronic pelvic pain—diagnostic laparoscopy and tubal ligation [26,27].

Acquisition of ultrasound guidance has also advanced the safety of regional anesthesia, especially because of the minimized risk of local anesthetic systemic toxicity after peripheral nerve blockade [26,27]. Monitoring of ultrasound decreased the risk of local anesthetic systemic toxicity after peripheral nerve blockade [28,29].

## 2. Materials and Methods

The authors reviewed abstracts and full-text articles concerning laparoscopic surgery, gynecology, anesthesia and COVID-19. Studies published in PubMed, Embase, Cochrane Library and found in Google Scholar before 1st February, 2021 were retrieved and analyzed.

SWOT analysis was used. It describes Strengths, Weaknesses, Opportunities and Threats and is mostly applied in strategic analysis. SWOT supports to examine internal and external factors influencing the topic of interest. SWOT analysis is a method, which can also be used in medicine. It supports the systematic integration of the patient and their individual issues into medical strategies [30]. In this review, SWOT analysis contributes to judge positive and negative aspects of regional anesthesia during COVID-19.

## 3. Results

### 3.1. Summary of Studies

A total of 83 studies were found concerning laparoscopy in regional anesthesia, all of them written in English. 17 studies including patient data were selected in gynecology and in general surgery about laparoscopy in regional anesthesia. In Covid time only one study compared laparoscopic surgery in general anesthesia to laparotomy and another study laparotomy in general anesthesia to regional anesthesia. Laparoscopy showed no disadvantage compared to laparotomy in Covid pandemic and in another study laparotomy in general anesthesia was associated with higher mortality and more pulmonary complications. In these studies, a maximum of 12 mm Hg CO_2_ pressure for pneumoperitoneum and a maximum of 15° Trendelenburg was used. Laparoscopic pelvic surgery in SA, Trendelenburg position of the patient was much higher, as much as 30–45° and this in combination with a low pneumoperitoneum pressure of ≤8. In 12 of 17 studies, in a small part of patients, additional intravenous sedation was given (Table 1).

Foremost, the review on gynecological laparoscopic surgeries (Table 2) provides evidence that degree of Trendelenburg position increased while the pressure of the pneumoperitoneum decreased.

### 3.2. SWOT Analysis

Collected review of papers help to categorize the available literature according to its strong and weak parameters whether it is internally or externally influenced. SWOT analysis guides the literature to classify into four parts (Table 3). Two parts (Strengths and Weaknesses) of the SWOT identifies the internal environment of the research focus while the remaining two parts (Opportunities and Threats) examine the external aspects of the research coverage.

With standardizing evidence-based SWOT four-part matrices for therapies, SWOT saves time in the decision-making process [79] and increases its precision [80,81].

## 4. Discussion

There is a recent change in the medical opinion based on new evidence, contradicting that in COVID-19 pandemic laparoscopic surgery should not be performed [5,6,7]. Laparoscopy has many advantages in favor of the protection against coronavirus transmission, if certain measures are undertaken. Laparoscopy in regional anesthesia prevents COVID-19, and this in in four ways: as a minimal invasive procedure it decreases the risk of virus contamination by blood, it decreases the risk of aerosol transmission through the upper airways, it results in shorter length of hospital stay compared with laparotomy and by this diminishes the risk of contamination and it spare the need for sedatives and hypnotics during this pandemic, when there are shortages of anesthetic drugs and intensive care material [6]. In the update of guidelines from the Society of American Gastroenterology and Endoscopic Surgeons and the European Association of Endoscopic Surgery stated due to the theoretical risk of Coronavirus transmission to health professionals: “Although previous research has shown that laparoscopy can lead to aerosolization of blood-borne viruses, there is no evidence to indicate that this effect is seen with COVID-19, nor would it be isolated to minimally invasive surgery (MIS) procedures” [8]. For virus transmission by generated smoke, laparoscopy is not more hazardous than open surgery.

This is enforced by the recent study of Yokoe et al., who examined Coronavirus in surgical smoke generated from tissue incised with an electrocautery scalpel [9]. Virus could be detected in the surgical smoke but was unable to induce infection in cultured cells. There it was observed that surgical masks decreased viral RNA detected by PCR to an amount of 99.80%. Therefore, smoke filtration by surgical masks provides sufficient protection against coronavirus, if other measures are respected such as smoke evacuation by aspiration during surgery and while removing trocars, performing surgery with low pneumoperitoneum pressure, and working precisely with an efficient and clean electrocautery scalpel [5,6]. Due to the small size of SARS-CoV-2, professional units capable to filtrate this virus efficiently from smoke are unfortunately commercially not available [5]. However, that is not necessary if simple measures are strictly followed [5,6]. However, to be on the safe side surgical interventions of COVID-19 patients should be performed only in emergency situations in specialized centers and elective surgeries postponed. It is debatable if asymptomatic Coronavirus positive patients should also be restricted to have surgery. It was reported that only symptomatic Coronavirus positive patients undergoing a surgical procedure were associated with morbidity and mortality [10]. There was also no difference observed in complications of patients having similar laparoscopic interventions in COVID pandemic compared to patients before the pandemic [11]. If precautions are taken by the health care personal and the indication is well checked, laparoscopic surgery combined with regional anesthesia may become the new standard as seen in cesarean section. Regional anesthesia for cesarean sections in COVID pandemic is advantageous for both the patients and the medical personal. It protects them from aerosol exposure and transmission of Coronavirus if general anesthesia with intubation is carried out. For the patient, it prevents respiratory problems seen with intubation and ventilation and possible deterioration of COVID-19 [12,13].

In the past the standard was to perform cesarean section in general anesthesia with intubation, whereas today regional anesthesia became standard for this operation. Clinical experiences from different international centers provides well-grounded evidence of laparoscopic surgery in regional anesthesia. There are two reasons why laparoscopic surgery compared to cesarean surgery is still carried out in general anesthesia. One is that laparoscopic surgery needs pneumoperitoneum and the other is that laparoscopic surgery for gynecology is mostly carried out in Trendelenburg. In Table 1 we summarized all publications concerning laparoscopy carried out in regional anesthesia. In pneumoperitoneum the pressure of 12 was not exceeded in any of the studies and most of them the pressure was under 10 mmHg. According to Jumaniyazov et al., laparoscopic surgery of the pelvis in 30–45° of Trendelenburg and SA is not only feasible but enables the surgeon to work comfortably with a pneumoperitoneum pressure ≤8. A degree of 30–45° Trendelenburg is well tolerated by the patient and allows to have a good vision of the pelvis anatomy even in low CO_2_ pressures. Such a low CO_2_ pressure in turn increases patient comfort and decreases the need for additional antalgic medicine. Such a procedure for COVID-19 infected patients, enables them to avoid complications and is an advantage in this pandemic.

### 4.1. Feasibility and Advantages to Perform Laparoscopy in Regional Anesthesia (RA) in COVID-19 Pandemic

The World Association of Anaesthetists and the Royal College of Anaesthetists recommend applying local or neuraxial regional anesthesia. Key anesthetic drugs required during the serious care of COVID-19 patients for such anesthesia are practicable and safe. Extra gained advantages of regional anesthesia during the COVID-19 pandemic are an absence of aerosol generating procedures (AGPs). This induces growing safety, economizing time and fewer resources needed as well as less financial costs for personal protective equipment (PPE). Compared to GA preservation of immune function, better postoperative analgesia, minimization of direct contact with medical personnel and early discharge are reported with RA [28]. Airway manipulation is linked with some of the highest percentages of COVID-19 transmission, and it is generally admitted to avoid Aerosol Generating Procedures (AGPs). RA decreases the risk of virus transmission from patient to the medical personnel [81,82,83]. Furthermore, RA enables excellent communication between patient, anesthesiologist and surgical team [84,85,86].

In addition to a reduction in pain and opioid use, postoperative pulmonary complications, postoperative nausea and vomiting, RA has been associated with a decrease in the incidence of postoperative cognitive dysfunction and delirium, which are common in GA [87,88].

In addition, RA has fewer effects on respiratory function and dynamics in comparison with GA, with or without inhibition of motor innervation of muscles [37,89,90]. The preservation of respiratory function decreases postoperative pulmonary complications in patients who may already have reduced respiratory function from COVID-19-related pneumonia or acute respiratory distress syndrome.

### 4.2. Minimal Risk of Laparoscopy during COVID-19

There is evidence regarding the greater risks of laparotomy versus minimally invasive surgery (MIS), specific to COVID-19 [7].

Even though previous research has shown that laparoscopy can lead to aerosolization of blood borne viruses, there is no evidence to indicate that this effect is seen with COVID-19 [40,41,91], if few precautions are taken in MIS procedures. For MIS procedures, use of devices to filter released CO_2_ for aerosolized particles should be considered. Proven benefits of laparoscopic surgery (LS) are of reduced length of stay and complications [43,44,45,46]. This should be considered along with the possibility for ultrafiltration of the main aerosolized particles. Filtration of aerosol particles during open surgery is more difficult than with laparoscopy [47,48,91]. However, care has to be taken from unexpected bursts of release of aerosolized particles mixed in CO_2_ from trocar valves during exchange of instruments. The transmission of COVID-19 during the removal of air and trocars may be the main reason. CO_2_ insufflation pressure should be balanced at a minimum and an ultrafiltration system (smoke evacuation system or filtration) should be utilized. The entire pneumoperitoneum should be safely vented via a filtration system before closure, trocar removal, specimen extraction or conversion to laparotomy [49].

### 4.3. Spinal Anesthesia in Laparoscopic Surgery for Gynecology

Patient position depends on the location to be operated—Trendelenburg position for the pelvic organs in gynecological procedures. Inducing spinal anesthesia by inexperienced personnel can be dangerous, because bupivacaine and other anesthesia in the spinal canal can block breathing. By using the described technique below and by monitoring and vigilance there is very low risk for the patient. In rare cases it may nevertheless happen, intubation material and devices for general anesthesia must be available for immediate use. The advantage of Trendelenburg is that it improves venous return and harmonizes blood pressure [27].

Some clinical trials verified that Peak Inspiratory Pressure (PIP) and plateau pressure climbed corresponding to intra-abdominal pressure increase in CO_2_ pneumoperitoneum, while the changes of the Trendelenburg position were not important. An increase in PIP will increase mean airway pressure and thus improve oxygenation. However, PIP may increase with any airway resistance such as increased secretions or bronchospasm. Thus, change of the position did not influence respiratory dynamics. The intra-abdominal pressure climb was presumably linked with pneumoperitoneum pushing of the diaphragm to the extent that Trendelenburg position cannot influence more on respiratory dynamics. Moreover, the pneumoperitoneum is expected to be more important to respiratory dynamics apart from position changes [24]. In clinical observations [24,25,26], lung compliance declined more after withdrawal of pneumoperitoneum than before pneumoperitoneum was installed. Other studies have mentioned that lung compliance recovered to baseline right after the pneumoperitoneum was withdrawn. Most probably, it is linked with insufficient time allowing retrieval of lung compliance and end-tidal carbon dioxide tension to baseline values after return to the supine position and elimination of CO_2_ from the peritoneal cavity. Furthermore, pulmonary function changes connected with surgery could not be removed. Pulmonary function is influenced by the types of operation, extent of intra-abdominal pressure and duration of pneumoperitoneum [24].

Laparoscopy will affect breathing of patients only in pneumoperitoneum with high pressure. This is why in all studies with laparoscopic surgery pneumoperitoneum was lower than 12 mmHg. In our experience of more than 1000 patients with laparoscopic surgery such low pressure is not affecting the respiratory function of the patients. In patients without contraindications, Trendelenburg is well tolerated by patients and in contrary to high pressure pneumoperitoneum, it has no negative effect on breathing of patients. Regional anesthesia has advantages such as minimal effect on the respiratory system, avoidance of intubation-related seeding of pathogens to the lower respiratory tract, decreasing thromboembolic complications and a reduced surgical stress response. Using regional anesthetic techniques, aerosol—generating procedures can be avoided with decreased risk to health personnel.

## 5. Conclusions

Based on the aforementioned analysis and studies performing laparoscopic surgery in spinal anesthesia and in Trendelenburg position, it is proposed that laparoscopic surgery in regional anesthesia is a reliable method during COVID-19 pandemic for operations. Regional anesthesia is a good alternative for laparoscopy and is well tolerated by patients, if carried out in low pressure pneumoperitoneum. If procedures are not urgent, surgery should be postponed in symptomatic Covid patients. Once recovered from COVID, a usual benefice/risk ration considering neuraxial anesthesia and general anesthesia is indicated. This is needed because more data are needed to confirm first data concerning higher mortality and pulmonary complications in patients with perioperative SARS-CoV-2 infection undergoing surgery in general anesthesia [43].

Further studies and analysis are needed to confirm the advantages of laparoscopic surgery in regional anesthesia.

## Figures and Tables

**Table 1 medicines-08-00060-t001:** Review of existing studies on regional anesthesia in general laparoscopic surgeries.

№	Authors	Total Number of Operations in Regional and General Anesthesia	Type of Anesthesia: SA = Spinal EA = Epidural CA = Combination of Spinal and Epidural GA = General	Type of Operation in Laparo- Scopy LC = Cholecyst-ectomy	Trendelen-Burg Position (Degrees) 0° = Supine Position	Pressure of Pneumo- Peritoneum (mmHg CO_2_)	Additional Antalgic Medicine: + = Yes - = No (Number of Patients)
1	Imbelloni et al., 2010 [31]	33	SA	LC	10°	8	+(10)
2	Singh 2015 [32]	50	SA	LC	0°	12	-
3	Gramatica et al., 2001 [16]	28	EA	LC	not mentioned	10	+
4	Ellakany 2013 [33]	40	SA	LC	Anti-Tren- delenburg	10	+(7)
5	Mehta 2010 [28]	60	SA-30 GA-30	LC	0°	12	-
6	Turkstani 2009 [27]	50	SA-25 GA-25	LC	Anti-Tren- delenburg	10	+
7	Tiwari 2013 [29]	224	SA-110 GA-114	LC	Trendelen- burg	8–10	+
8	Shahriari et al., 2015 [34]	80	SA	hernia repair	0°	not mentioned	+
9	Sinha [35]	4099	SA	Abdominal urologic	not mentioned	8–10	+

Source: developed by authors.

**Table 2 medicines-08-00060-t002:** Review of existing studies on regional anesthesia in gynecological laparoscopic surgeries.

№	Authors	Total Number of Operations in Regional and General Anesthesia	Type of Anesthesia: SA = Spinal EA = Epidural CA = Combination of Spinal and EpiduralGA = General	Type of Operation in Laparo- Scopy LC = Cholecyst-ectomy	Trendelen- Burg Position (Degrees) 0° = Supine Position	Pressure of Pneumo-Peritoneum (mmHg CO_2_)	Additional Antalgic Medicine: + = Yes - = No (Number of Patients)
1	Giampaolino 2019 [36]	1	Regional Anesthesia (not specify)	laparosc. removal of IUD during pregnancy	12°	8	not mentioned
2	Chauvet 2020 [37]	1	CA	adnexectomy	10–15°	6–8	-
3	Uzman 2017 [38]	33	CA	appendec-tomy	15°	10	-
4	Raimondo 2020 [25]	13	SA	ovarian cyst- ectomy, adnex-ectomy	Minimal Trendelen- burg	9.7 ± 2.1	+(1)
5	Asgari 2017 [39]	56	SA	infertility	0°	8–10	+
6	Pusapati 2010 [40]	41	SA	Gynaeco- logy	10° (lithotomy position)	6 ± 4.47	+(3)
7	Moawad 2018 [41]	1	SA	total laparo- scopic hyster- ectomy	15°	12	+
8	Sinha 2008 [35]	514	SA	cyst- ectomy and diagnostic laparoscopy	Not mentioned	8–10	+
9	Jumaniyazov 2021 [42]	912	SA	Gynaeco- logy (adnexe, adherence, myomectomy, infertility, EUG)	30–45° Trendelen- burg position (lithotomy position)	≤ 8	+(17)

Source: developed by authors.

**Table 3 medicines-08-00060-t003:** SWOT analysis on regional anesthesia during COVID-19.

Strengths	Weakness (Counter Indication)
(1) No aerosol generating manipulation; Reducing cough during intubation and extubation—reducing the possibility of transmission of COVID [43,44]; (2) Reducing the contact of the anesthesiologist with the upper respiratory tract [45]; (3) No need for special equipment (virus filter, negative pressure operating unit) [46,47]; (4) Reducing the amount of drugs used, which are in small quantities during a pandemic [39]; (5) No pulmonary complications [43,44]; (6) Reducing the load on the lungs with early pneumonia [45]; (7) Reducing the duration of postoperative pain [46]; (8) Reducing the use of opioids [47]; (9) Reducing vomiting and nausea [48]; (10) Lack of cognitive dysfunction and delirium [49]. In severe comorbid patients less morbidity and mortality outcomes [50]; (11) Short hospital days [51]; (12) Rapid restoration of gastrointestinal function [52];	1. The absolute contraindications Shock and bleeding in emergency [53]; keeping the patient in horizontal position takes around 10 min that delays the operation in urgent situations [54]; intracranial pressure (ICP) [47,49]
	2. Relative contraindications are: Pre Existing neurological disease (e.g., multiple sclerosis) [38] Severe dehydration (hypovolemia) [50] age greater than 50 years [18,39] Obesity and chronic hypertension [55]; Thrombocytopenia or coagulopathy [56,57]; severe mitral and aortic stenosis [57] left ventricular outflow obstruction as observed with hypertrophic obstructive cardiomyopathy [18]
	3. Impact on Trendelenburg: (a) for gynecology: Trendelenburg position increases intracranial pressure and may inhibit breathing [58]; (b) Trendelenburg position is contraindicated in reduced right ventricular function (RVEF) and chronic obstructive pulmonary disease (COPD) [59];
Opportunity	Threats (consequences, complications)
(1) Decrease in the frequency of cardiovascular diseases [60]; (2) Reducing the incidence of pulmonary complications [59]; (3) Prevention of deep vein thrombosis and pulmonary embolism [60,61]; (4) minimize the negative impact of intra-abdominal overpressure on various organs and systems [60]; (5) Provides rejection of myorelaxants [55,62]; (6) The relative stability of hemodynamic parameters and cerebral blood flow under regional anesthesia is associated with the absence of the inhibitory effect of mechanical ventilation, the preservation of the tone of the diaphragm, which prevents an increase in transthoracic pressure transmitted from the abdominal cavity [3]; (7) there is no significant decrease in the vital capacity of the lungs and the minute ventilation volume throughout the entire intra- and postoperative period when using epidural anesthesia [13]; (8) preservation of spontaneous breathing during laparoscopic operations in the Trendelenburg position ensures adequate pulmonary ventilation, and hemoglobin oxygen saturation is 96–98% [19]; (9) the possibility of prolonging the epidural block in the postoperative period [19,63]; (10) the advantage of EA is a long-term analgesic effect (up to 6 h) [4]; (11) Can be applied with contra indication parameters to general anesthesia [64,65]; (12) 12) Trendelenburg position increases venous return and cardiac output consequently increasing the perfusion of organs [66,67];	(1) hypotension [63,64] myocardial depression, [59,65], bradycardia [66,67] (2) shoulder pain [68] (3) limited intra-abdominal pressure [69,70] and decreased view of intra-abdominal organs [71,72] (4) diaphragmatic irritation [73] (5) discomfort [59] with nausea [74] and vomiting [8] (6) feeling of lack of air (although hemodynamics are stable [75], saturation is normal) (due to pneumoperitoneum and intestinal mobilization towards the diaphragm) [76] (7) Spinal block [77] (8) Backache [78] (9) Postdural puncture headache [37,79]

Source: developed by authors.

## Data Availability

Not applicable.

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
