# Peer review of "Laparoscopy in Gynecologic and Abdominal Surgery in Regional (Spinal, Peridural) Anesthesia, the Utility of the Technique during COVID-19 Pandemic"

_medicines, 2021, doi:10.3390/medicines8100060_

Round 1

Reviewer 1 Report

You did a lot of work reviewing the literature concerning anesthesia in laparoscopic surgery. But you have to change your aim. It is not possible to make any conclusions about laparoscopy, anesthesia and covid-19 without even one study in your review with patients infected sars-cov2. 

Author Response

Dear Reviewer,

Many thanks for sharing your valuable suggestions to improve the paper. Authors seriously worked on that and providing the following changes to the paper. We would highly appreciate constructive approach from your side to make this paper publishable. For any questions, we are ready to respond. Please, find our detailed responses below.

Best regards,

Authors team

Reviewer 2 Report

The authors developed the case promoting neuraxial anesthesia for laparoscopic procedure  in covid 19 patients

Major concerns

While this is almost  true and I agree with them,   they should specify only for  emergency surgery since even when surgery is necessary except for extreme urgency most surgery can be postponed until symptoms and PCR  test are negative and in this cases we return back to usual case of balancing positive and negative drawbacks of neuraxial  vs general anesthesia in these context which is not new 

of course cesarean section and most emergency surgery can be applied in this context of pandemic 

Line 47 please describe Pfannenstiel incision

I  don't understand why studies including  combined general and regional anesthesia  are included  table 1 they should be removed 

Swot analysis 

failure of neuraxial anesthesia should be stated 

this failure can be a failure in puncture but also a failure providing anesthesia which can lead to emergency intubation procedure with additional risk for covid 19 propagation 

the conclusion should  state that if procedures are not urgent surgery should be postoponed until recovery other wise this is a usual benefice /risk ration considering neuraxial anesthesia and general anesthesia 

Author Response

(The authors gave the same response as above.)

Round 2

Reviewer 1 Report

If you insist.

The title

Your review of the literature does not concerns patients infected with Sars-cov 2, your title is misleading, something weaker will be appropriate, for example: "Laparoscopy in gynecologic and abdominal surgery in regional (spinal, peridural) anesthesia, the utility of the technique during Covid-19 pandemic."

Introduction

point 1

"However due to complications caused by Covid infection general anesthesia with intubation should be avoided in non- specialized centers. If an emergency situation requires surgery, it should therefore be performed in neuraxial regional anesthesia like spinal or peridural anesthesia"  -please make an appropriate citation, the second statement is controversial it means that the superiority of regional versus general anesthesia is proved. 

point 2

"The feasibility of laparoscopic surgery in regional anesthesia was demonstrated and published" - please make an appropriate citation

point 3

"There is a recent change in the medical opinion based on new evidence, contradicting that in COVID-19 pandemic laparoscopic surgery should not be performed [5, 6, 7]. Laparoscopy has many advantages in favor of the protection against coronavirus transmission, if certain measures are undertaken. Laparoscopy in regional anesthesia prevents COVID-19, and this in in four ways: as a minimal invasive procedure it decreases the risk of virus contamination by blood, it decreases the risk of aerosol transmission through the upper airways, it results in shorter length of hospital stay compared with laparotomy and by this diminishes the risk of contamination and it spare the need for sedatives and hypnotics during this pandemic, when there are shortages of anesthetic drugs and intensive care material [6]. In the update of guidelines from the Society of American Gastroenterology and Endoscopic Surgeons and the European Association of Endoscopic Surgery stated due to the theoretical risk of Coronavirus transmission to health professionals: “Although previous research has shown that laparoscopy can lead to aerosolization of blood-borne viruses, there is no evidence to indicate that this effect is seen with COVID-19, nor would it be isolated to minimally invasive surgery (MIS) procedures” [8]. For virus transmission by generated smoke, laparoscopy is not more hazardous than open surgery.This is enforced by the recent study of Yokoe & al, who examined Coronavirus in surgical smoke generated from tissue incised with an electrocautery scalpel [9]. Virus could be detected in the surgical smoke but was unable to induce infection in cultured cells. There it was observed that surgical masks decreased viral RNA detected by PCR to an amount of 99.80%. Therefore smoke filtration by surgical masks provides sufficient protection against coronavirus, if other measures are respected like smoke evacuation by aspiration during surgery and while removing trocars, performing surgery with low pneumoperitoneum pressure, and working precisely with an efficient and clean electrocautery scalpel [5, 6]. Because of the small size of SARS-CoV-2, professional units capable to filtrate this virus efficiently from smoke are unfortunately commercially not available [5]. But that’s not necessary if simple measures are strictly followed [5, 6]. However, to be on the safe side surgical interventions of COVID-19 patients should be performed only in emergency situations in specialized centers and elective surgeries postponed. It is debatable if asymptomatic Coronavirus positive patients should also be restricted to have surgery. It was reported that only symptomatic Coronavirus positive patients undergoing a surgical procedure were associated with morbidity and mortality [10]. There was also no difference observed in complications of patients having similar laparoscopic interventions in COVID pandemic compared to patients before the pandemic [11]. If precautions are taken by the health care personal and the indication is well checked, laparoscopic surgery combined with regional anesthesia may become the new standard as seen in cesarean section. Regional anesthesia for cesarean sections in COVID pandemic is advantageous for both the patients and the medical personal. It protects them from aerosol exposure and transmission of Coronavirus if general anesthesia with intubation is carried out. For the patient, it prevents respiratory problems seen with intubation and ventilation and possible deterioration of COVID-19 [12, 13]."

this is a very good text and I think should be a part of the discussion.

point 4

"The aim of this review is to analyze the studies on laparoscopy in regional anesthesia before and during COVID pandemic and to take some conclusions concerning the feasibility" - I think your point is:

"The aim of this review is to analyze the studies on laparoscopy in regional anesthesia and to take some conclusions concerning its utility  in Covid pandemic"

please place it at the end of the introduction

2. Materials and methods

point 5

"SWOT analysis was used. It describes Strengths, Weaknesses, Opportunities and Threats and is mostly applied in strategic analysis. SWOT supports to examine internal and external factors influencing the topic of interest. SWOT analysis is a method, which can also be used in medicine. It supports the systematic integration of the patient and their individual issues into medical strategies." - please make an appropriate citation

Results

Point 6

"A total of 83 studies were found concerning this topic, all of them written in English. 17 studies including patient data were selected in gynecology and in general surgery about laparoscopy in regional anesthesia, however none described Covid-19 status." - please describe the remaining studies

 point 7

section 3.2. table3 - there are weaknesses without citation - explain

point 8

Please remove the following -

"3.3. Advantages of performing laparoscopy in regional anesthesia during COVID infection.

In the literature, we found recommendations and studies doing laparoscopic surgery under regional anesthesia during Covid-19 pandemic, but without Covid-19 status. Referring to the literature data, in particular, the recommendations of scientists about regional anesthesia, as well as laparoscopy in patients during Covid-19 pandemic and based on our own experience, where we analyzed the results of laparoscopic surgery in spinal anesthesia, gives us the opportunity to recommend the use of laparoscopic surgery in regional anesthesia in patients with SARS-COVID-2."

this section is unclear, the information is given earlier and the rest is written in the discussion.

Discussion

point 9

"There is evidence regarding the greater risks of laparotomy versus minimally invasive surgery (MIS), specific to COVID-19."-please make appropriate citation

Conclusions

point 10

"Based on the aforementioned analysis and studies performing laparoscopic surgery in spinal anesthesia and in Trendelenburg position, it is proposed that laparoscopic surgery in regional anesthesia is a reliable method during COVID-19 pandemic for operations not lasting more than 2-3 hours. - the last condition "2-3 hours" was not mentioned earlier. Conclusion have to emerge from your study/review- please remove "not lasting more than 2-3 hours"

point 11

"Laparoscopy in regional anesthesia in Covid time is to be preferred in countries in which the necessary material and infrastructure is not available for prevention of Covid infection" - why? you did not discuss this earlier, there is nothing about this in the results

Point 12

"Once recovered from COVID, a usual benefice/risk ration considering neuraxial anesthesia and general anesthesia is indicated." - please clarify the statement

Abstract

point 13

"Patients infected with Covid-19 needing surgery are at risk for Covid-19 complications in general anesthesia in contrast to regional anesthesia. Proved long-term complications of Covid-19 (3 months – 1 year) were reported to be: fibrosis of the lungs, bronchitis, chronic pulmonary failure, myocarditis, arrhythmias and chronic heart failure, liver and kidney damage, demyelination of nerve fibers, cognitive impairment, depression and psychosis." - abstract should reflect the study. You do not have this information in the introduction - please correct

point 14

"3 topics concerning laparoscopic surgery were reviewed: 1) Achievements in laparoscopy; 2) Advantages of regional anesthesia compared to general anesthesia; 3) Feasibility to perform laparoscopy in regional anesthesia in Covid-19 pandemic." - please clarify whether this is an introduction or methodology

point 15

"In Covid time only one study compared laparoscopic surgery in general anesthesia to laparotomy and another study laparotomy in general anesthesia to regional anesthesia. Laparoscopy showed no disadvantage compared to laparotomy in Covid pandemic and in another study laparotomy in general anesthesia was associated with higher mortality and more pulmonary complications."- please move this text to results

point 16

please match conclusions in your study with conclusions in the abstract, they should be the same

And the abstract should match the text

Author Response

Dear Reviewer,

The authors provided immediate response to your respectful comments. Please, find below one by one responses to your comments please. If any questions, we remain at your disposal. 

Reviewer 2 Report

The authors have responded to most of my queries 

please include the conclusion in the abstract  ( about emergency surgery)

Author Response

Dear Reviewer,

The authors provided immediate response to your respectful comment. Please, find below our response to your comment please. If any questions, we remain at your disposal. 

Round 3

Reviewer 1 Report

point 15 

"In Covid time only one study compared laparoscopic surgery in general anesthesia to laparotomy and another study laparotomy in general anesthesia to regional anesthesia. Laparoscopy showed no disadvantage compared to laparotomy in Covid pandemic and in another study laparotomy in general anesthesia was associated with higher mortality and more pulmonary complications."- please place this text in results  3.1., not only in abstract.

Author Response

Dear Reviewer,

The authors provided immediate responses to your respectful comments. Please, find below our response to your comment please:

1) Authors updated the results and added the text as you suggested. Now, the results part is enriched with the selected text.

This manuscript is a resubmission of an earlier submission. The following is a list of the peer review reports and author responses from that submission.

Round 1

Reviewer 1 Report

The authors reviewed abstracts and full-text articles concerning following key words: laparoscopic surgery, gynecology, anesthesia and Covid-19. Studies published in PubMed, Embase, Cochrane Library and found in Google Scholar before 1st February, 2021 were analyzed. They finally analyzed 17 studies including patient data that were selected in gynecology and in general surgery about laparoscopy in regional anesthesia, however none described Covid-19 status.

First of all, the explanations received by the authors according to my comments from the previous review are premature.

The authors performed technical modification only (for example, they split two tables into one) or performed just minor modification to the main text, but the main major problems of this review still remains the same. The point was to investigate laparoscopy in gynecologic and abdominal surgery in regional (spinal, peridural) anesthesia during Covid-19 pandemic but none of the included studies does not describe COVID-19 status. How it is possible to conclude anything if baseline parameters were not matched with the aims of the study.

Beside this, the authors should learn how to perform systematic review. Methodology is poor, below every standard. Many important data are missing, there is no inclusion / exclusion criteria, strategy of the literature review, keywords, primary / secondary outcomes….)

The conclusions drawn from the study are just general statements, not following from the results.

Reviewer 2 Report

I understand authors' clear-cut explanation of safety and feasibility of laparoscopy under pneumoperitoneum with low pressure. One concern is that these successful cases were procedures in patients possessing near-normal respiratory and circulatory function. The authors have to show and discuss cases with impaired respiratory and circulatory function because respiratory function and corresponding circulatory function of COVID-19 patients was frequently impaired even when COVID-19 was successfully treated.